# Probiotic-Loaded Bacterial Cellulose as an Alternative to Combat Carbapenem-Resistant Bacterial Infections

**DOI:** 10.3390/antibiotics13111003

**Published:** 2024-10-25

**Authors:** José Gutiérrez-Fernández, Laura Cerezo-Collado, Víctor Garcés, Pablo Alarcón-Guijo, José M. Delgado-López, Jose M. Dominguez-Vera

**Affiliations:** 1Department of Microbiology, Virgen de las Nieves University Hospital, 18014 Granada, Spain; 2Departmento de Química Inorgánica, Instituto de Biotecnología, Universidad de Granada. 18071 Granada, Spain; lauracerezo@ugr.es (L.C.-C.); vigaro@ugr.es (V.G.); palarcon@ugr.es (P.A.-G.)

**Keywords:** resistant bacteria, probiotics, living material, bacterial cellulose

## Abstract

**Background:** Carbapenems are one of the mainstays of treatment for antibiotic-resistant bacteria (ARB). This has made the rise of carbapenem-resistant bacteria a threat to global health. In fact, the World Health Organization (WHO) has identified carbapenem-resistant bacteria as critical pathogens, and the development of novel antibacterials capable of combating infections caused by these bacteria is a priority. **Objective**: With the aim of finding new alternatives to fight against ARB and especially against carbapenem-resistant bacteria, we have developed a series of living materials formed by incorporating the probiotics *Lactobacillus plantarum* (*Lp*), *Lactobacillus fermentum* (*Lf*), and a mixture of both (*L. plantarum*+*L. fermentum*) into bacterial cellulose (BC). **Results**: These probiotic-loaded bacterial celluloses inhibited the proliferation of three ARB, including two carbapenem-resistant enterobacteria (CRE), identified as *Klebsiella pneumoniae* and *Enterobacter cloacae*, and a carbapenem-resistant *Pseudomonas aeruginosa*. Interestingly, while the probiotics *L. plantarum*, *L. fermentum*, and the mixture of both were found to be inactive against these ARB, they became active once incorporated into BC. **Conclusions**: The increase in activity is due to the known effect that cells increase their activity once incorporated into a suitable matrix, forming a living material. For the same reasons, the probiotics in the living materials BC–*L. plantarum*, BC–*L. fermentum*, and BC–*L. plantarum*+*L. fermentum* showed increased stability, allowing them to be stored with bacterial activity for long periods of time (two months).

## 1. Introduction

Antibiotic-resistant bacteria (ARB) are a significant global health problem. Recent reports indicate that these bacteria will be associated with nearly 5 million deaths in 2019 [1]. Infections caused by ARB do not respond effectively to standard treatments, making them difficult or impossible to treat and increasing the risk of serious complications and death. The newest and most effective defense against these infections is the carbapenem antibiotics, a subgroup of β-lactams. These drugs target penicillin-binding proteins that are critical for peptidoglycan synthesis, disrupting cell wall formation and leading to bacterial cell lysis. However, bacteria are becoming increasingly resistant to these therapies, making it necessary to combine carbapenem antibiotics with β-lactamase inhibitors to effectively treat ARB infections. β-lactamase inhibitors are enzymes located in the periplasmic space that can effectively neutralize the action of most β-lactam antibiotics [2,3]. This has led to a sharp increase in the use of carbapenem antibiotics, but also to the emergence of infections that are becoming resistant to them [4].

This situation seems to indicate the imminent demise of conventional antibiotics, which has led to an urgent need to identify new drugs that can replace them. Scientists are investigating unconventional approaches to ARB treatment, including nanotechnology, photodynamic and photothermal therapy, and probiotics. With respect to nanotechnology, the interest in the use of nanoparticles is based on the fact that their small size and large specific surface area facilitate increased contact with bacterial biofilms, a central target of ARB, thereby providing an opportunity to enhance their antibacterial efficacy [5]. However, because nanoparticles can have deleterious toxicological effects on the human body, they are not commonly used in clinical practice. In addition, photosensitizers with aggregation-induced emission are also being tested [6]. These agents adhere to bacteria and, after irradiation, induce aggregation and increase reactive oxygen species, which are highly toxic to bacteria and can induce their death. However, the main limitation of this approach is that it is mainly effective for topical infections where penetration of radiation is not a problem. Another alternative to the use of antibiotics is probiotics. Probiotics are microorganisms expected to confer health benefits. They produce antimicrobial compounds and are therefore candidates for the prevention of CRE [7] and treatment of ARB [8,9]. However, probiotics are vulnerable to the hostile environment of infected tissue, which jeopardizes their viability and consequently their applications against ARB [10]. In the food industry, probiotics have been encapsulated in various matrices to preserve their viability and protect them from the low pH and bile present in the gastrointestinal tract. Examples of these matrices include polysaccharides such as cellulose, chitosan and alginate, proteins such as gelatin, collagen and casein, and lipids such as natural waxes and glycerol [10,11].

Given the lack of truly effective treatments against ARB, this study presents the development of an antibiotic-free therapy using bacteria of the genus *Lactobacillus* encapsulated in the bacterial cellulose matrix, forming probiotic-loaded bacterial celluloses. In particular, we chose *L. plantarum* and *L. fermentum* because they are considered two of the probiotics with the broadest spectrum of antibacterial activity, with inhibitory activity against gram-negative and gram-positive bacteria [12,13]. They produce various antimicrobial compounds such as organic acids, hydrogen peroxide, and antimicrobial peptides such as bacteriocins. In addition, both are used as food preservatives and have been proposed as potential alternatives to antibiotics [14,15].

Inhibition trials were conducted to demonstrate how these probiotic-loaded bacterial celluloses can combat Gram-negative bacteria resistant to carbapenem antibiotics. The carbapenem-resistant Gram-negative bacteria studied included carbapenem-resistant *Enterobacteriaceae* (CRE) [16] (*Enterobacter cloacae* and *Klebsiella pneumoniae*) and carbapenem-resistant *Pseudomonas aeruginosa*. Together with *Acinetobacter baumannii*, these bacteria are recognized by the WHO as priority clinical pathogens [4]. *K. pneumoniae* is an opportunistic pathogenic gram-negative bacterium and represents the most prevalent strain of CRE, often associated with significant outbreaks in healthcare facilities [4]. This bacterium is capable of causing a wide range of infectious diseases, including pneumonia, urinary tract infections (UTIs), bacteremia, and liver abscesses [17]. The clinical challenge posed by *K. pneumoniae* stems from its widespread global spread through mobile genetic elements, resulting in significantly higher mortality rates compared to non-carbapenem-resistant strains [18]. *E. cloacae*, akin to *K. pneumoniae*, is a frequently encountered microorganism in clinical isolates, contributing to various infections such as pneumonia, UTIs, and sepsis [19]. Within the *Enterobacter* genus, *E. cloacae* ranks as the third most common bacterium implicated in nosocomial infections [20]. On the other hand, *P. aeruginosa* is the leading cause of nosocomial infections by gram-negative bacteria, and it is the leading agent causing mortality and morbidity in patients with cystic fibrosis [21]. Infections stemming from *P. aeruginosa* are marked by escalating mortality rates and heightened resistance to carbapenems. Consequently, there is an urgent need to explore alternative antimicrobial strategies given its propensity for resistance not only to carbapenems but also to other antibiotics within the group of β-lactams, including novel β-lactamase inhibitor therapies [22].

The approach that we have used to address the challenge of finding antibiotic-free alternatives for the treatment of CRE was inspired by an emerging class of synthetic materials called living materials, which are formed by integrating living biological entities into an inert matrix. It has been shown that in living materials, the abiotic component enhances the stability and activity of the living entity [23]. With the aim of finding new alternatives to fight against ARB and especially against carbapenem-resistant bacteria, we report here the encapsulation of *L. plantarum* (*Lp*), *L. fermentum* (*Lf*), and their mixture (*L. plantarum*+*L. fermentum*) in bacterial cellulose (BC) to form probiotic-loaded bacterial celluloses. These living materials consist of a nonliving matrix of cellulose nanofibers in which living and active *L. plantarum, L. fermentum*, or the mixture of both are perfectly integrated [24]. We have exploited the beneficial effect of encapsulation, and we report here the efficacy of the probiotic-loaded bacterial celluloses BC–*L. plantarum*, BC–*L. fermentum*, and BC–*L. plantarum*+*L. fermentum* as a novel and potent strategy to combat CRE.

## 2. Results and Discussion

Probiotics are live microorganisms that provide health benefits to the host, either by restoring the natural balance of bacteria in the microbiota or by secreting antipathogenic compounds [25]. The key to using probiotics against infection is that the probiotics must colonize the infected tissue and remain alive to release active bactericidal species. However, the chemical scenario of bacterial infection is not optimal for their proliferation. Thus, when we examined the effect of the two probiotics *L. plantarum* and *L. fermentum* and their mixture (*L. plantarum*+*L. fermentum*) against antibiotic-resistant *E. cloacae* in optimal media for pathogens, we found no effect. As shown in Figure 1, no inhibition rings were observed in the agar diffusion tests for any probiotic against *E. cloacae*. These results indicate that when *E. cloacae* and the probiotic (*L. plantarum*, *L. fermentum*, or *L. plantarum*+*L. fermentum*) meet in an environment that is optimal for the pathogens and not for the probiotics, which is a realistic scenario of a real infection, no bacterial inhibition occurs.

We then turned to encapsulating the probiotics to assess whether, once encapsulated, they could withstand the hostile environment imposed by the pathogen, which would thus allow their proliferation and activity. We then encapsulated the two probiotics *L. plantarum* and *L. fermentum* and a mixture of both in BC, following a procedure developed by our group [24]. Using this protocol, the probiotics are fully integrated throughout the cellulose matrix, as shown in the Field Emission Scanning Electron Microscopy (FESEM) images of BC–*L. plantarum*, BC–*L. fermentum*, and BC–*L. plantarum*+*L. fermentum* (Figure 2). The fact that all the bacteria in BC are encapsulated makes it possible to compare the activity of free and fully encapsulated probiotics, allowing us to evaluate the effect of encapsulation on the antibacterial activity of probiotics.

We identified the *Lactobacillus* species in the samples using ASV (amplicon sequence variant) analysis (see Section 3.4). The V3–V4 hypervariable regions were sequenced using 16S rRNA sequencing because this method provides accurate taxonomic resolution and is widely used in microbiota research, often via Illumina MiSeq sequencing technology [26]. This approach allowed us to distinguish and determine the relative abundances of *L. fermentum* and *L. plantarum*. The resolved ASVs and their assignment to *Lactobacillus* species are shown in Appendix A. This analysis, together with the qPCR results, showed that the absolute abundances of ASVs of *Lactobacillus* in the samples were as follows: BC–*L. plantarum*, 67.7% (and 32.3% of *K. xylinus*); BC–*L. fermentum*, 81% (and 19% of *K. xylinus*); and BC–*L. plantarum*+*L. fermentum*, 81.9% (53.2% of *L. fermentum*, 28.7% of *L. plantarum*, and 18.1% of *K. xylinus*).

After the negative results with free probiotics (Figure 1), we tested the antibacterial activity of BC–*L. plantarum*+*L. fermentum*, BC–*L. plantarum*, and BC–*L. fermentum* against the ARB *E. cloacae*, *K. pneumoniae*, and *P. aeruginosa*. The agar diffusion experiments are shown in Figure 3, in which the ARB were grown in tryptic soy broth agar (TSA), and the three encapsulated samples produced inhibition zones against the three ARB. These results demonstrated that the encapsulated probiotic of BC–*L. plantarum*+*L. fermentum*, BC–*L. plantarum*, and BC–*L. fermentum* inhibited the proliferation of the ARB. The greater activity of encapsulated probiotics compared to free probiotics is due to the synergy that exists between the living and nonliving blocks of a living material [23]. In materials such as these probiotic-loaded bacterial celluloses, in addition to protection, the inert matrix (BC) may provide an optimal medium for biofilm formation. BC promotes the attachment of probiotics to the surface and to each other, leading to colonization and biofilm formation. In line with this, it is interesting to note that other different bacterial cellulose-containing *bacillus* species have been reported to have antimicrobial activity [27,28], but none against resistant bacteria, let alone carbapenem-resistant bacteria. These materials were produced using an adsorption–incubation method where the bacillus only penetrates a few layers of the bacterial cellulose. In BC–*L. plantarum*, BC–*L. fermentum*, and BC–*L. plantarum*+*L. fermentum*, however, the *lactobacilli* are actually entrapped in the cellulose matrix, where colonization and biofilm formation are more favorable. This may explain the higher antimicrobial efficacy of probiotic-loaded bacterial celluloses compared to these other cellulose *bacilli*.

To confirm that the antimicrobial activity of BC–*L. plantarum*+*L. fermentum*, BC–*L. plantarum*, and BC–*L. fermentum* was due to the encapsulated probiotic and not to the cellulose-producing bacterium *Komagataeibacter xylinus* (*K. xylinus*), inhibition studies were performed, which showed that the BC–*K. xylinus* sample, i.e., native bacterial cellulose without bacterial elimination, had no antimicrobial activity, not even against a nonresistant *P. aeruginosa* strain (Appendix A). Therefore, the comparison of the results with free or integrated probiotics in BC confirms the enhanced activity of probiotics once they are integrated in BC, and allows us to postulate the use of probiotic-containing BC materials as an alternative in the fight against ARB, including CRE.

On the other hand, due to the effect of increasing the stability of probiotics once they are incorporated into a living material, the probiotics of BC–*L. plantarum* and BC–*L. fermentum* offer the advantage of being storable for a considerable time without loss of usability. An indirect approach to determine the stability of these probiotics is to measure the evolution of the pH over time, since these (lactic acid) probiotics secrete lactic acid, which sets the pH at around 4 (pKa = 3.80). We therefore measured the evolution of pH of BC–*L. plantarum* and BC–*L. fermentum* over time (Figure 4).

Preservation results after one and two months of storage at 4 °C showed continued lactic acid secretion with pH versus time (Figure 4), with the typical pattern of live and active lactic acid bacteria. For BC–*L. fermentum*, no differences were observed between fresh, freeze-dried, and non-freeze-dried samples after one month of storage (Figure 4A). By contrast, after two months of storage (Figure 4B), the freeze-dried BC–*L. fermentum* more closely resembled freshly synthesized probiotic cellulose, reaching a pH of approximately 4 after 4.5 h. A similar pattern was observed for BC–*L. plantarum*, although only the freeze-dried samples retained fermentative capacity after two months of storage, while the non-lyophilized samples failed to acidify the medium. The decrease in pH is due to the live and active probiotics in BC–*L. plantarum* and BC–*L. fermentum*, as neither pure BC nor BC–*K. xylinus* reduced the pH below 5.8 after 24 h (Appendix A).

These results show that the encapsulation of these probiotics in BC not only affects the bacterial inhibitory activity but also the stability over time. BC–*L. plantarum* and BC–*L. fermentum* can be stored for one month without the need for lyophilization. Once lyophilized, both show activity after two months, even though lyophilization was performed without the addition of cryoprotectants, indicating that the BC matrix acts in some way as a cryoprotectant for the probiotics it contains.

## 3. Materials and Methods

### 3.1. Reagents

Reagents used in this study, NaCl, NaOH, HCl, K_2_HPO_4_, KH_2_PO_4_, HNa_2_PO_4_, cellulase from *Trichoderma reesei*, tryptic soy broth, glucose, peptone from meat, yeast extract, citric acid monohydrate, and agar, were acquired from Sigma-Aldrich (Darmstadt, Germany). Ultrapure water (18.2 MW/cm, bacteria < 0.1 CFU/mL at 25 °C, Milli-Q, Millipore, Burlington, MA, USA) was used for the preparation of aqueous solutions.

### 3.2. Bacterial Strains

The cellulose-producing bacteria belonging to the species *K. xylinus* (CECT 473, *K. xylinus*) were provided by the Colección Española de Cultivos Tipo (CECT, Paterna, Spain) in a lyophilized form, cultured in Hestrin–Schramn (HS) [29] agar medium at 30 °C, and preserved freeze-dried at −4 °C. The probiotic bacterial strain *Lactobacillus plantarum* (CECT 220, *Lp*) was supplied by CECT, and *Lactobacillus fermentum* (*Lf*) was provided by Biosearch Life S.A. (Madrid, Spain). All were grown in Man, Rogosa, and Sharpe medium (MRS, SICAL) at 37 °C and stored freeze-dried at −4 °C.

CRE *E. cloacae* (carbapenemase-type resistance VIM-1+SHV-12; ST78) and *K. pneumoniae* (carbapenemase-type resistance OXA-48+CTX-M-15+SHV-28+TEM-1; ST307), as well as carbapenem-resistant *P. aeruginosa* (carbapenemase-type resistance VIM-2; ST233, cgST3407), were obtained from clinical urine cultures at the Microbiology Laboratory of the Virgen de las Nieves University Hospital (HUVN) (Granada, Spain). Carbapenemases were detected with the colorimetric Neo-Rapid CARB Kit (Rosco Diagnostica A/S, Taastrup, Denmark) and using immunochromatography (NG5-Test Carba, NG Biotech, Guipry, France for KPC, NDM, VIM, IMP, and OXA-48-like enzymes) for isolates with values above EUCAST breakpoints for carbapenemase-producing bacteria screening. The carbapenemase-producing type was confirmed by the Andalusian Laboratory of Molecular Typification of the Spanish Program for the Prevention and Control of Healthcare-related Infections and Appropriate Utilization of Antibiotics (acronym in Spanish, PIRASOA) via massive sequencing (Illumina Inc., San Diego, CA, USA) using CLC GenomicsWorkbench v10 (Qiagen, Hilden, Germany), ResFinder (Lyngby, Denmark) (http://genepi.food.dtu.dk/resfinder), and CARD databases (Hamilton, ON, Canada) (https://card.mcmaster.ca). All clinical bacterial isolates were cultured in TSB at 37 °C and preserved in the cold room of the HUVN Microbiology laboratory.

### 3.3. Encapsulation of Probiotics in BC. Synthesis of BC–L. plantarum, BC–L. fermentum, and BC–L. plantarum+L. fermentum

Probiotic-containing bacterial celluloses were prepared according to a previously reported protocol [24]. Briefly, 0.1 mL of *K. xylinus* suspension (OD600 nm = 0.3) was mixed with 0.1 mL (corresponding to 1.069 × 10^8^ CFU/mL for *L. plantarum* and 1.696 × 10^8^ CFU/mL for *L. fermentum*) of the corresponding probiotic bacteria *L. plantarum*, *L. fermentum*, or a mixture of both in a 1:1 ratio in 1 mL of HS medium, and these bacteria were co-cultured under aerobic conditions at 30 °C for 3 days to produce a thick cellulose membrane at the liquid–air interface. The HS medium was replaced with MRS, and the cellulose membranes were incubated under anaerobic conditions at 37 °C for 48 h and replaced with fresh medium after 24 h.

### 3.4. Identification of Probiotics in BC–L. plantarum, BC–L. fermentum, and BC–L. plantarum+L. fermentum

Identifying the probiotic or probiotic mixture in BC was carried out by Microomics Systems S.L. (Barcelona, Spain) as follows:

1. Library preparation and sequencing. Samples were amplified using 16S rRNA V3–V4 region-specific primers (forward, 5′-TCGTCGGCAGCGTCAGATGTGTATAAGAGACAGCCTACGGGNGGCWGCAG-3′; reverse, 5′-GTCTCGTGGGCTCGGAGATGTGTATAAGAGACAGGACTACHVGGGTATCTAATCC-3′). Amplification was carried out after 25 PCR cycles. Positive (CM) and negative (CN) controls were used for quality control. The positive control is a mock community control and was processed in the same way as the samples. Sequencing was performed on an Illumina MiSeq with 2 × 300 bp reads. In addition, the DNA used to construct the sequencing libraries was quantified using the qPCR results of the 16S rRNA V3–V4 region. The 16S rRNA copy data were used to normalize the sequencing data.

2. Amplicon sequence processing and analysis. Raw demultiplexed forward and reverse reads were processed using the following methods and pipelines as implemented in the software QIIME2 version 2020.11 with default parameters unless otherwise stated [30]. DADA2 software (https://github.com/benjjneb/dada2) was used for quality filtering, denoising, pair-end merging, and amplicon sequence variant calling (ASV, i.e., phylotypes) using qiime dada2 denoise-paired method [31]. Quality threshold was established to define read sizes for trimming before merging. Reads were truncated at the position where the 75th percentile Phred score was felt to be below the established threshold (<Q20).

ASVs were aligned using the qiime alignment mafft method [32]. The alignment was used to build a tree and calculate phylogenetic relationships between ASVs using the qiime phylogeny fasttree method [33]. ASV tables were subsampled without replacement to balance sample sizes for diversity analysis using the qiime diversity core-metrics-phylogenetic pipeline. Phylotype data were used to calculate the following alpha diversity metrics: richness and Pielou’s evenness. The phylotype and phylogenetic data were used to calculate beta diversity and unweighted and weighted Unifrac, Jaccard, and Bray Curtis distances. Taxonomic assignment of ASVs was performed via a Bayesian classifier trained on the SILVA v138 database (i.e., 99% OTUs database) [34] using the qiime feature-classifier classify-sklearn method. The *Lactobacillus* genus was reassigned using BLAST v2.12 against an in-house filtered SILVA database. To understand the differences between the *Lactobacillus* V3–V4 sequences, a phylogenetic tree was constructed using all *Lactobacillus* genus sequences from the SILVA database. Sequence alignment was performed with MUSCLE 5 (Multiple Sequence Comparison by Log-Expectation) [35], and the phylogenetic tree was generated using RAxML version 8, with the best tree model selected by the algorithm. Some ASVs had identical statistics from the BLAST search, making it difficult to assign a species to the given ASVs. In these cases, the species was assigned manually if one of the used species was among the best hits.

The relative abundances of each phylotype were normalized to the qPCR data using the following formula:$$\mathrm{ASV}\;s\;\mathrm{Absolute}\;\mathrm{abundace}=\frac{\mathrm{read}\;\mathrm{ASV}}{\mathrm{coverage}\;\mathrm{sample}}*\frac{\mathrm{qPCR}\;\mathrm{counts}}{\mathrm{ng}}*\frac{\mathrm{sample}\;\mathrm{concentration}}{\mu l}*4\mu l\;1^{a}\;\mathrm{PCR}\;\frac{1}{\mathrm{sample}\;g}$$

### 3.5. Field Emission Scanning Electron Microscopy (FESEM)

The samples BC–*L. plantarum*, BC–*L. fermentum*, and BC–*L. plantarum*+*L. fermentum* were dehydrated with ethanol using a concentration gradient (30%, 50%, 70%, 96%, and 100% dry; 5 min each step, except for 100%, which was performed twice). Then, they were introduced into hexamethyldisilazane according to a concentration gradient (1%, 10%, 25%, 50%, and 100%, 10 min each step). The samples were left to dry at room temperature overnight. Subsequently, the dehydrated samples were mounted on pieces of aluminum using carbon tape and analyzed using an FESEM (Zeiss SUPRA40V, Jena, Germany) located in the Scientific Instrumentation Center (University of Granada, CIC-UGR).

### 3.6. Study of the Activity of BC–L. fermentum and BC–L. plantarum

The probiotic activity of BC*–L. fermentum* and BC*–L. plantarum* was evaluated after their storage at 4 °C for 1 month and 2 months, and in both lyophilized and non-lyophilized states, and comparisons between freshly synthesized (time 0) bacteria, bacteria-free cellulose BC, and unpurified bacterial cellulose BC*–K. Xylinus* were made. For freeze-drying, samples were frozen by placing them in liquid nitrogen, and they were immediately lyophilized using a Unifreez FD-8 lyophilizer (DAIHAN Scientific Co., Wonju, Republic of Korea). After these storage times, BC*–L. fermentum* and BC*–L. plantarum* were added to 10 mL of MRS broth media, incubated at 37 °C, and 180 rpm for 6 h. Aliquots of 0.5 mL were collected sequentially every 1 h from the supernatants of each culture. The lactic acid production of BC*–L. fermentum*, BC*–L. plantarum*, BC, and BC*–K. Xylinus* was monitored by pH measurements of the aliquots [36].

### 3.7. Inhibitory Activity Against ARB

To compare the antibacterial activity of free and encapsulated probiotics, triplicate samples of the probiotic celluloses BC–*L. plantarum*, BC–*L. fermentum*, and BC–*L. plantarum*+*L. fermentum,* and the probiotic samples *L. plantarum*, *L. fermentum*, and *L. plantarum*+*L. fermentum,* which were obtained after cellulase digestion of the corresponding probiotic-containing celluloses, were tested for their activity against carbapenem-resistant *P. aeruginosa* as well as carbapenem-resistant *E. cloacae* and *K. pneumoniae* using agar diffusion in TSA [24], a favorable medium for the growth of these pathogenic microorganisms [22]. A total of 0.1 mL of a suspension of the pathogenic bacteria (OD600 nm = 0.3) was homogeneously spread on petri dishes containing TSA. The probiotic celluloses BC–*L. plantarum*, BC–*L. fermentum*, and BC–*L. plantarum*+*L. fermentum* or the corresponding free probiotics *L. plantarum*, *L. fermentum*, *and L. plantarum*+*L. fermentum* were then placed on TSA plates inoculated with the pathogen. These cells were incubated for 24 h at 37 °C, and the presence of inhibition halos was verified and imaged.

## 4. Conclusions

We have shown that the encapsulation of *L. plantarum* and *L. fermentum* in bacterial cellulose enhances their activity. Thus, while these free probiotics were not active against the carbapenem-resistant *K. pneumoniae*, *E. cloacae*, and *P. aeruginosa*, the probiotic-loaded bacterial celluloses formed by incorporating these probiotics into bacterial cellulose made them active. In addition, the improved stability of the probiotics once they were incorporated into the bacterial cellulose allowed them to be easily stored with preserved bacterial activity for long periods of time (at least two months). These results are particularly relevant, as the WHO has identified carbapenem-resistant bacteria as critical pathogens. In addition, these results point to a promising way to combat carbapenem-resistant bacterial infections: the encapsulation of probiotics, which are not effective in their free form but become active when encapsulated in a suitable matrix.

## Figures and Tables

**Figure 1 antibiotics-13-01003-f001:**
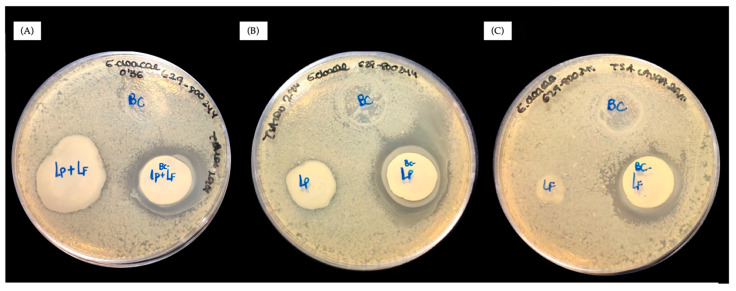
Agar diffusion assays of free and encapsulated *L. plantarum*+*L. fermentum* (**A**), *L. plantarum* (**B**) and *L. fermentum* (**C**) on tryptic soy broth agar (TSA) plates containing antibiotic-resistant *E. cloacae*.

**Figure 2 antibiotics-13-01003-f002:**
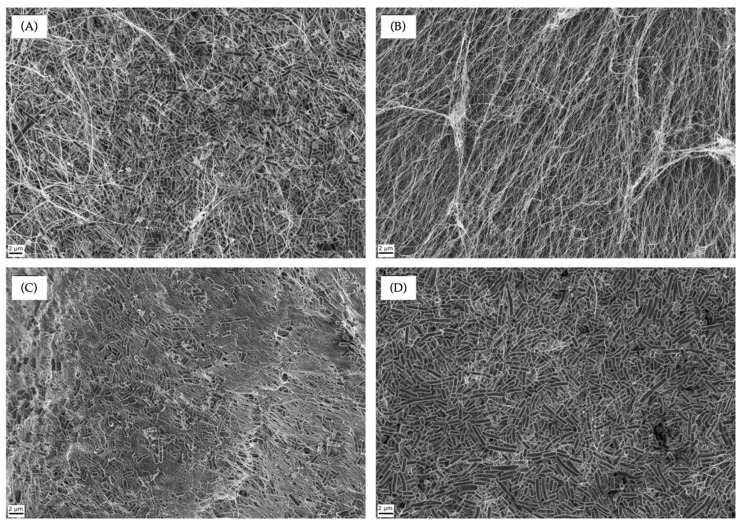
FESEM micrographs of (**A**) BC–*L. plantarum*+*L. fermentum*; (**B**) pure BC; (**C**) BC–*L. fermentum*; and (**D**) BC–*L. plantarum*.

**Figure 3 antibiotics-13-01003-f003:**
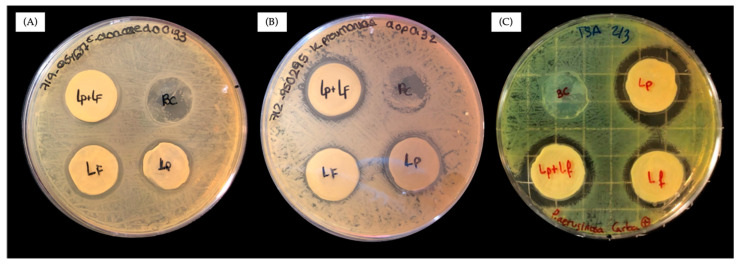
Inhibitory activity of BC–*L. plantarum*, BC–*L. fermentum*, and BC–*L. plantarum*+*L. fermentum* against (**A**) *E. cloacae*; (**B**) *K. pneumoniae*; and (**C**) *P. aeruginosa*.

**Figure 4 antibiotics-13-01003-f004:**
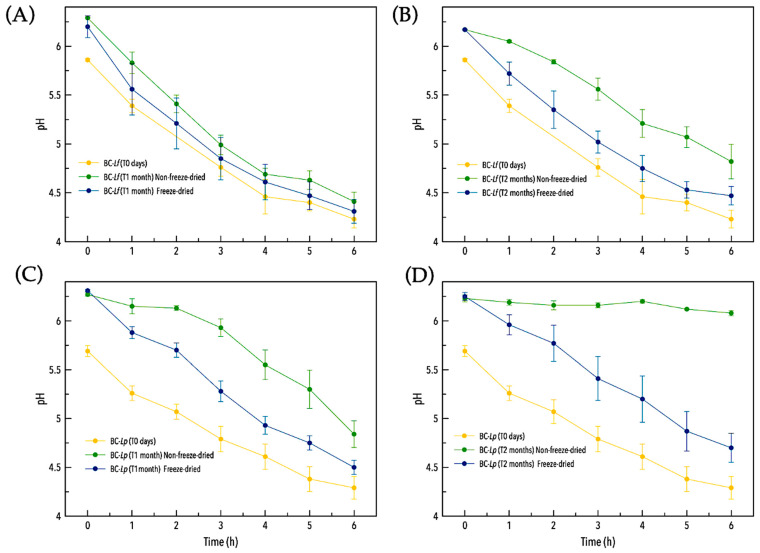
Evolution of pH in MRS of BC–*L. fermentum* after one (**A**) and two (**B**) months of storage at 4 °C. Each panel shows a comparison between freshly prepared BC–*L. fermentum* (t0 days), non-freeze-dried samples, and freeze-dried samples. The same is shown for BC–*L. plantarum* after one (**C**) and two months (**D**).

## Data Availability

The original contributions presented in this study are included in the article and Appendix A; further inquiries can be directed to the corresponding authors.

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
