# Peer review of "Probiotic-Loaded Bacterial Cellulose as an Alternative to Combat Carbapenem-Resistant Bacterial Infections"

_antibiotics, 2024, doi:10.3390/antibiotics13111003_

Round 1
Reviewer 1 Report
Comments and Suggestions for Authors
The manuscript describes on Probiotic-based Living Material as an Alternative to Combat Car- 2
bapenem-resistant Bacterial Infections. Overall, the topic of manuscript is of interest but the data presentation need to be improved. There were inconsistency throughout the whole manuscript (e.g., title, method, result). There were unclear for the statement indicating the Probiotic-based Living Material or encapsuled probiotics in the experiment used in throughout the maunscript. The definition needs to be revised for throughout the whole manuscript. Data presentation and discussion were weak. It’s not the right way to use the abbreviation (LP, LF) for stating on the name of microorganism. Please revise the name of microorganism to L. plantarum or L. fermentum. The authors only reported the results and lack of the explanation provided. I would suggest trying to strengthen the sentences. The result from the Amplicon sequences processing was missing. The sentences were hard to follow for the reader. The native speaker for lauguage check is required. The typing error must be rechecked. The manuscript in the current version can not be considered for publishing. I encourage strengthen the manuscript and provide more novelty result not only the Inhibitory activity but include more Amplicon sequences processing and microbiome.
Specific comments
Title: the use of Probiotic-based Living Material is too wide and not indicated to the bacterial cellulose (BC), while the authou used only BC as a probiotic carrier. I would reccommnd to include the bacterial cellulose (BC) in the title and be more specific.
Abstact: the objective of the study was missing. I recommend to rewritten.
Line 12: it’s not the right way to use the abbreviation for stating on the name of microorganism.
The authors may use the abbreviation only in the table or figure, but it should not be in the pargraph.
Line 15-18: “Ahough the probiotics Lp, Lf and the mixture of both, Lp+Lf, were found to be inactive against these ARB, they became active once incorporated into BC. The increase in activity is due to the known effect that cells increase their activity once incorporated into a suitable matrix, forming a living material”
Please provide short explanation why active once incorporated into BC. What is the criteria of a suitable matrix forming a living material
Introduction part:
The introduction is too short. Please provide the example of experiments on probiotic or other nanomaterial for caring probiotics. I would suggest trying to strengthen the sentences.
Line 14 and 30: please select the use the same abbreviaiton (ARB vs ARBs)
Line 32-36: I recommend to rewritten. The sentences were hard to follow. The statement was not clear on the efficiency and treatment for ARBs
Line 45-47: “a central target of ARBs, thereby providing an opportunity to enhance their antibacterial efficacy.”
The BC seems to increase the ability to withstand surrouding enviroments and livable more, not the direct effect to increase their antibacterial efficacy
Line 53-58: please remove the Phage therapy, which did not reflect to this experiment
Line 64-70: “This study demonstrates the efficacy of encapsulated probiotics Lactobacillus planta- 64 rum (Lp), Lactobacillus fermentum (Lf) and the mixture of both (Lp+Lf) in bacterial cellulose 65 (BC) against PA resistant to carbapenem antibiotics expressing VIM-2 resistance and two 66 carbapenem-resistant Enterobacteriaceae (CRE) strains”
These sentences sound like the conclusion of this study. Why did the autthor put them in the introduction.
Line 72-83: “KP is an opportunistic pathogenic gram-negative bacterium “
What did KP, EC,PA mean?
Please recheck Grammar error
Please revise the name of microorganism
Please provide the detail on how many strains were used and the purpose of each strain use for. The results should be based on the statistical analysis with showing the significant difference.
Line 89-96: Please provide the objective and hypotheses of the study.
Avoid using the word “living materials” and be more specific to BC.
Result and Discussion
I recommend to rewritten. Too many general theories provided. It sounds like few discussions and explanation. Please consider link the results and make discussion. I would suggest trying to strengthen the sentences. There were many tpying error. Please recheck the manuscript. The results should report as the number and base on statistical analysis.
Please revise the name of microorganism
Line 118: “We then turned to encapsulating the probiotics to assess whether, once encapsulated”
No information on the encapsulation in the meterial and methodolody.
Line 125: “The abundances of Lactobacillus in BC-Lp, BC-Lf and BC-Lp+Lf were 124 calculated by DNA sequencing (see Materials and Methods)”
Please reconsider to show the result (figure or table) obtained from the DNA sequencing. It will be more interesting for the paper.
The author showed only the result of the numebr of bacteria obtained from qPCR , but the result from DNA sequencing was missing.
Please be careful and make clarification that the author encapsulated or innnoculated the probiotics in the BC.
Line 157: “Preservation results after one and two months of storage at 4oC”
The time points for evaluation of number and bacterial population were too wide. It was hard to identify the change in microorganism.
Materials and methods
Many points were unclear and there was inconsistency with the result
(eg., no result on DNA sequencing in the result part)
Line 176: “All high-grade reagents”
Please clarify what reagents.
Line 182: “Hestrin–Schramn (HS) agar medium”
Please include (company, city, country) after the medium or chemical used
Please recheck throughout the whole Materials and methods
Line 183-185: “served at -4oC”
Please recheck the temperature.
4.4. Identification of probiotics in Bc-Lp+Lf, BC-Lf and BC-Lp.
1. Library preparation and sequencing.
2. Amplicon sequences processing and analysis.
Please add the reference and add the result from Amplicon sequences processing
Line 253-254: “The abundances of Lactobacillus in the samples were: BC-Lp, 67.7% and 32.3% of Kx; BC-Lf, 81% and 19% of Kx and BC-Lf+Lp, 81.9% (53.2% of Lf and 28.7% of Lp) and 18.1% of Kx.
Why did the author report the result in the MM part.
Conclusion
Please rewrite the conclusion to be more novelty and sumarize all important results from each factor or each experiment.
Figures
Figure 1. Agar diffusion assays on TSA plates containing antibiotic-resistant EC. Inhibition zones 113 were not found around the free probiotics (samples on the left side of each plate): (A) Lp+Lf; (B) Lp; 114 (C) Lf. However, inhibition zones were evident when the probiotics (Lp or Lf) or their mixture (Lp+Lf) 115 were encapsulated in BC (samples on the right side of each plate). The cellulose matrix (BC) showed 116 any inhibition effect (top of each plate).
Please avoid to put the result in the figure legend. Please remove these sentences
Please revise the figure legend of all figure.
Comments on the Quality of English LanguageExtensive editing of English language required.
Reviewer 2 Report
Comments and Suggestions for Authors
In this research article, authors used probiotic bacteria, Lactobacillus plantarum (Lp), Lactobacillus fermentum (Lf), and the mixture of both (Lp+Lf) into bacterial cellulose (BC), and studied the proliferation of three antibiotic-resistant bacteria, including two carbapenem-resistant enterobacteria. These probiotics inhibited the proliferation of antibiotic-resistant bacteria, including two carbapenem-resistant enterobacteria. This research paper is very interesting to read and showed some interesting result.
Title of the manuscript is too general, it should be specific to this work.
Introduction: In the introduction please add recent references. The introduction should be hypothesis driven. So, please do required changes. The novelty of the paper must be clearly mentioned in the introduction.
In the abstract, please include background of the study in the first line.
Line 78: KP stems? Please check
The aim of the study must be clearer in the introduction section.
Material and methods are clear and very interesting. The authors prepared probiotic-containing bacterial celluloses for the determination of antimicrobial activity.
Line 267: Is it is lactic acid?. or organic acids??.. Any spectrophotometry or HPLC assays performed for the determination of lactic acid? Please clarify.
Line 278: 0.1 ml of a suspension of the patho- genic bacteria (OD600nm= 0.3) was homogeneously spread on petri dishes containing.. Please use cell density (CFU/mL) or log CFU? If possible
In the discussion, in depth analysis required to compare the present study with previous report on antimicrobial activity of Lactobacillus-cellulose materials. Discussion is not sufficient. Please include 4-5 recent references to support your findings.
References are not according to the journal format. Please format carefully.
Reviewer 3 Report
Comments and Suggestions for Authors
Comments for the authors:
The manuscript presents an investigation into the use of probiotic-based living materials to combat carbapenem-resistant bacterial infections. This study is highly relevant, considering the urgent global challenge posed by antimicrobial resistance (AMR), particularly carbapenem-resistant organisms (CROs).
The study offers critical insights into the efficacy of probiotics encapsulated in bacterial cellulose (BC) against three resistant bacterial strains. Below, I have outlined some questions and suggestions for the authors’ consideration:
1. I noticed that several abbreviations, like TSA, and Kx, are used frequently but aren't always defined when they first appear. Terms like SILVA and MUSCLE 5 that might be clear to experts, should be explicitly defined. This would make the manuscript more accessible to readers who might not be as familiar with the specific terminology.
2. While the general methods for encapsulating probiotics in BC are described, more explanation on the choice of bacterial cellulose as the encapsulation matrix would help justify the approach.
3. The rationale for selecting Lactobacillus plantarum and Lactobacillus fermentum as the probiotics, as well as the specific bacterial strains tested, is not entirely clear. Providing more context or justification for these choices (e.g., their clinical relevance or prevalence in resistant infections) would strengthen the manuscript’s experimental design.
4. This manuscript has some typo and grammar errors. The authors should modify them very carefully. For example:
In line 87, change “group of b-lactams, including novel b-lactamase inhibitor therapies” to “group of β-lactams, including novel β-lactamase inhibitor therapies”.
In summary, this manuscript presents valuable insights into the field of antimicrobial alternatives, specifically the use of probiotic-based living materials against carbapenem-resistant bacteria. The research is timely and important, given the increasing prevalence of antibiotic resistance. While the findings are robust and relevant, several revisions related to clarity, reproducibility, and minor typographical errors could improve the manuscript’s accessibility and overall impact.
Round 2
Reviewer 1 Report
Comments and Suggestions for Authors
Comments to Author:
The manuscript describes “Probiotic-loaded Bacterial Cellulose as an Alternative to Combat 2
Carbapenem-resistant Bacterial Infections”. Thank you for the revision accordingly. The manuscript was much improved. However, there are still major revision in the way to write the name of bacteria in the results and discussion part and the heading of the methodology. Do not use the abbreviation of BC-Lp, BC-Lf, BC-Lf+Lp for the results and discussion. Also, the author only reported the results. Only little explanation and discussion were provided. Please add the discussion. The typo needs to be checked throughout the manuscript.
Specific comments
Line 104: please remove “and”
Line 102-110: check front
Line 133-134: “BC-Lp, BC-Lf and BC-Lp+Lf”
please provide the full detail of bacterial name
line 135: please explain the ASV analysis before using the abbreviation
line 143-144, line 149, line 153, 163, 167, 183, 184-193:
Please provide the full detail of bacterial name and treatment. Do not use the abbreviation of BC-Lp, BC-Lf, BC-Lf+Lp for writing the results. It can use in the figures.
Line 113-144: the author only reported the results. Please add the discussion.
Line 178-197: the author only reported the results. Please add the discussion.
Figure 1 Figure 2 Figure 3, figure 4:
Please provide these information in the figure legends
L. plantarum (Lp) or L. fermentum (Lf) L. or plantarum+L. fermentum (Lf+Lp)
Figure 4: please provide the information for the comparison of non-freeze dried vs freeze dried under the figure legend.
Method
Line 203: “NaCl, NaOH, HCl, K2HPO4, KH2PO4, HNa2PO4, cellulase”
Please add (company name , city, country) for each chemical reagent.
4.3 Encapsulation of probiotics in BC. Synthesis of BC-Lp, Bc-Lf and BC-Lp+Lf.
4.4. Identification of probiotics in Bc-Lp+Lf, BC-Lf and BC-Lp. 2
Please provide the full detail of bacterial name and treatment.
Line 263-270: please recheck the word “ASVs” Some ASVs word were italic
Round 3
Reviewer 1 Report
Comments and Suggestions for Authors
The manuscript can be accepted for publishing.